# The HLA-DRB1*07 Allele Is Associated with Interstitial Lung Abnormalities (ILA) and Subpleural Location in a Mexican Mestizo Population

**DOI:** 10.3390/biom12111662

**Published:** 2022-11-09

**Authors:** Ivette Buendia-Roldan, Marco Antonio Ponce-Gallegos, Daniela Lara-Beltrán, Alma D. Del Ángel-Pablo, Gloria Pérez-Rubio, Mayra Mejía, Moises Selman, Ramcés Falfán-Valencia

**Affiliations:** 1Translational Research Laboratory on Aging and Pulmonary Fibrosis, Instituto Nacional de Enfermedades Respiratorias Ismael Cosio Villegas, Mexico City 14080, Mexico; 2HLA Laboratory, Instituto Nacional de Enfermedades Respiratorias Ismael Cosio Villegas, Mexico City 14080, Mexico; 3Interstitial Lung Disease and Rheumatology Unit, Instituto Nacional de Enfermedades Respiratorias Ismael Cosio Villegas, Mexico City 14080, Mexico

**Keywords:** HLA DRB1*07, ILA, subpleural ILA, central ILA

## Abstract

Interstitial lung abnormalities (ILA) are defined as the presence of different patterns of increased lung density, including ground glass attenuation and reticular opacities on chest high-resolution computed tomography (HRCT). In this study, we included 90 subjects with ILA and 189 healthy controls (HC) from our Aging Lung Program. We found that subjects with ILA are older, have a significant smoking history, and have worse pulmonary function than HC (*p* < 0.05). When we evaluated the allele frequencies of the human leukocyte antigen (HLA) system, we found that HLA-DRB1*07 was associated with a higher risk for ILA (*p* < 0.05, OR = 1.95, 95% CI = 1.06–3.57). When we compared subjects with subpleural ILA vs. HC, the association with HLA-DRB1*07 became stronger than the whole ILA group (*p* < 0.05, OR = 2.29, 95% CI = 1.24–4.25). Furthermore, subjects with subpleural ILA and central ILA display differences in allele frequencies with HLA-DRB1*14 (3.33% vs. 13.33%, *p* < 0.05) and *15 (3.33% vs. 20%, *p* < 0.05). Our findings indicate that the HLA-DRB1*07 allele contributes to the risk of ILA, especially those of subpleural locations.

## 1. Introduction

Interstitial lung abnormalities (ILA) are a recently emerged concept regarding incidental findings of subtle interstitial abnormalities on chest high-resolution computed tomography (HRCT). ILA are defined as the nondependent presence of different patterns of increased lung density, including ground glass attenuation and reticular opacities affecting more than 5% extent of a lung zone. Once such abnormalities have been recognized, the distribution patterns, such as central (non-subpleural), subpleural nonfibrotic, and subpleural fibrotic ILA, are relevant, because they may have different prognostic implications [1]. To date, the estimated prevalence of ILA in subjects older than 60 is 7% in nonsmokers and around 10% in smokers [2,3,4,5]. Importantly, ILA may have several adverse outcomes, including progressive functional decline, the development of progressive fibrosis, increased lung cancer risk, hospitalization, and all-cause mortality, strongly indicating a potential clinical significance [6,7].

Age and smoking are cardinal demographic risk factors for ILA, and in terms of genetic susceptibility, the gain-of-function rs35705950 promoter polymorphism of the *MUC5B* gene is strongly associated with ILA [8]. In addition, ILA has been linked with several gene variants previously reported in IPF [9]. Supporting the role of these genetic associations, it has been shown that ILA is more frequent in relatives of patients with both familial and sporadic pulmonary fibrosis than in the general population [10].

However, to our knowledge, no previous studies raised the question of to what extent the human leukocyte antigen (HLA) system can be a critical part of the host genetic factors contributing to ILA susceptibility and prognosis.

The HLA system plays a key role in antigen presentation and T-cell activation by displaying thousands of peptides at T-cell receptors for self-/non-self-discrimination, and many of its gene variants have been implicated in disease associations [11]. Previous studies have revealed that different HLA classes and alleles are associated with diverse interstitial lung diseases. For example, HLA-DRB1*15 and some HLA-A and -B alleles were more frequent in IPF patients than in healthy controls. Similarly, HLA-DRB1*04 and some alleles of the ancestral haplotype (HLA-DRB1*03:01-DQB1*02:01) are associated with hypersensitivity pneumonitis (HP) [12,13,14,15]. Moreover, via genome-wide genotype imputation association analyses, two strongly linked HLA alleles were found to be associated with fibrotic idiopathic interstitial pneumonia HLA-DRB1*15:01 and -DQB1*06:02, affecting the expression of HLA genes in lung tissue and indicating that the potential genetic risk due to HLA alleles may involve gene regulation in addition to an altered protein structure [16].

In this study, we aimed to evaluate the participation of HLA class II (HLA-DRB1 loci) in the genetic susceptibility for ILA and the different patterns of ILA subtypes in a Mexican mestizo population cohort.

## 2. Materials and Methods

From March 2015 to July 2019, almost 900 respiratory asymptomatic volunteers >60 years old living in Mexico City for at least 5 years were recruited into the “Lung Aging Program” of the Instituto Nacional de Enfermedades Respiratorias Ismael Cosío Villegas (INER). From these enrolled subjects, we identified and included in this study 90 subjects that presented interstitial lung abnormalities (ILA), which are defined as asymptomatic respiratory individuals that showed by high-resolution computed tomography (HRCT) mainly the presence of ground glass attenuation (GGA), diffuse nodules (both associated with inflammatory ILA), reticular opacities (RO), honeycombing, or traction bronchiectasis (associated with fibrotic ILA) affecting more than 5% of the total lung. Additionally, 189 healthy controls (HC) with normal HRCT were randomly selected from the same cohort and were included as a control group. Afterward, we divided the ILA group according to the HRCT pattern: ground glass attenuation, reticular opacities, and subpleural and central localization for further analysis, agreeing with the Fleischner Society [1].

At the first visit, in addition to the HRCT, forced vital capacity (FVC), forced expiratory volume in the 1st second (FEV_1_), diffusing capacity of the lung for carbon monoxide (DL_CO_), and the six-minute walking test (oxygen desaturation and 6-MW distance) were measured in all controls and ILA subjects, in accordance with the relevant guidelines and regulations. Additionally, we applied questionnaires for demographic data and obtained blood samples. All participants signed a consent letter, and the Scientific and Ethical Committee of INER, Mexico, approved this study (C76–17 and B14–17).

### 2.1. HLA Genotyping

The DNA was extracted from peripheral blood cells via venipuncture using the commercial BDtract Genomic DNA isolation kit (Maxim Biotech, San Francisco, CA, USA), according to the manufacturers’ instructions. After that, the DNA concentration was assessed by UV absorption spectrophotometry at the 260-nm wavelength via a NanoDrop device (Thermo Scientific, Wilmington, DE, USA).

HLA class II (HLA-DR) low-resolution genotyping was achieved through polymerase chain reaction (PCR) with sequence-specific primers (SSP), using Micro SSPTM Generic HLA Class II DNA Typing Tray–DRB Only Lot 004 (One Lambda, SSP2LB, Canoga Park, CA, USA). The HLA-DRB typing includes alleles in the DRB1, DRB3, DRB4, and DRB5 loci in 24 independent reactions. The DRB loci allele group covers DRB1*01, *03, *04, *07, *08, *11, *12, *13, *14, *15, and *16, as well as DRB3*02, DRB3*03, DRB4*01, and DRB5*01 specificities (DR51, DR52, and DR53 serological equivalents).

Taq DNA polymerase recombinant (Ref. EP0402. Thermo Scientific. Vilnius, Lithuania) was employed in all PCR amplifications. The amplicon products were electrophoresed in 2% agarose gels containing 0.2 µg/mL ethidium bromide (cat. E1385, Sigma-Aldrich, Schnelldorf, Germany) for 40 min (30 V/cm), and amplified bands were visualized in a dual-intensity UV light transilluminator (UVP Inc., Upland, CA, USA) before being stored and evaluated in the Electrophoresis Documentation and Analysis System (EDAS 290) (Eastman Kodak, New Haven, CT, USA). A ladder 100-bp molecular weight marker (cat. CSL-MDNA-100BPH, Cleaver Scientific, United Kingdom) was employed to facilitate the allele-specific weight band identification.

### 2.2. Statistical Analysis

Descriptive data were presented as frequency tables, median, and minimum and maximum values. We performed a normality test with Kolmogorov–Smirnov, and a comparison between groups was performed by the chi-square and Fisher’s exact tests for categorical values and Mann–Whitney *U* Test for continuous variables. Values of *p* < 0.05 were considered significant. Allele frequencies were determined by direct counting from alleles identified in each participant, and the differences between groups were evaluated by determining and comparing the allele frequencies through 2 × 2 contingency tables. Statistical significance was assessed using Epi Info 7.1.4.0 statistical software (CDC, Atlanta, GA, USA), considering the χ^2^ value to compare case and control groups. The results were considered significant when the *p*-value was <0.05 and corrected by Yate’s test; similarly, the odds ratios (OR) with 95% confidence intervals (CI) were estimated to determine the strength of the association. After that, the logistic regression analysis was performed to adjust for possible confounding variables (age and smoking).

Analyses were performed using IBM SPSS Statistics for Mac OO v. 25.0 (Armonk, NY: IBM Corp., NY, USA) and R studio.

## 3. Results

### 3.1. Demographic and Clinical Characteristics

Ninety ILA subjects and 189 HC were included in the study. We found that ILA subjects were older than HC (<0.05), and a major percentage of ILA subjects were smokers (61.11% vs. 47.31%) with more years of smoking history (25 vs. 12.5) compared to HC subjects. Although, within normal limits, ILA subjects displayed worse pulmonary function, including fewer walked meters during the 6-min walking test (6-MWT), lower oxygen saturation (SaO_2_) at rest and after exercise, and lower FVC and DL_CO_ than HC subjects (*p* < 0.05). Additionally, when we compared the subpleural ILA group vs. HC, differences in age, initial and final SaO_2_, and DL_CO_ were still maintained (*p* < 0.05). The complete data are shown in Table 1.

A comparison of subpleural ILA with central ILA only revealed that those subjects with central ILA had lower FVC (Appendix A).

### 3.2. Allele Frequencies between ILA and HC Subjects

We identified 13 alleles in both groups. Interestingly, we found that the HLA-DRB1*07 allele was more frequent in the ILA group compared with HC (12.22% vs. 6.64%, *p* =0.02, OR = 1.95, CI 95% = 1.06–3.57), and the association was maintained after Yates’ correction (*p* = 0.04). The results are shown in Table 2. In addition, when we compared the other genes in the HLA kit, we found that HLA-DR53 was more prevalent in the ILA group than in the HC group (56.47% vs. 45%), although it did not reach statistical significance. The results are shown in Appendix A.

### 3.3. Logistic Regression Analysis in the ILA and HC Groups

A logistic regression analysis was performed to adjust possible confounding covariables (age and smoking). HLA-DRB1*07 maintained its association with ILA (*p* = 0.037; OR = 2.11; 95% CI = 1.04–4.28). In addition, age, smoking, and the intercept also demonstrated a significant association (*p* < 0.05) (Table 3).

### 3.4. Allele Frequencies between Subpleural ILA and HC Subjects

We divided the ILA group according to the abnormality location in the lung parenchyma (subpleural and central). In an intra-case comparison between subpleural vs. central ILA, we found that HLA-DRB1*14 and -DRB1*15 were more frequent in the central ILA group (13.33% vs. 3.33% and 20% vs. 3.33%, respectively; *p* < 0.05). The allele frequencies are shown in Table 4. After that, we compared the allele frequencies between subpleural ILA vs. HC, finding that the HLA-DRB1*07 allele was associated with a higher risk for subpleural ILA (14% vs. 6.61%, *p* = 0.007; OR = 2.29; 95% CI = 1.24–4.25). This association was maintained after Yates’ correction (*p* = 0.011). The results are shown in Table 5.

### 3.5. Logistic Regression Analysis in the Subpleural ILA and HC Groups

A logistic regression analysis was performed, adjusted by age. Remarkably, HLA-DRB1*07 maintained its association with subpleural ILA (*p* = 0.009; OR = 2.615; 95% CI = 1.26–5.28). In addition, age and the intercept demonstrated a significant association (*p* < 0.05) (Table 6).

### 3.6. Allele Frequencies between ILA HRCT Patterns and HC Subjects

We divided the ILA group according to two different HRCT patterns: GGA and RO and compared the allele frequencies between them. Interestingly, we found that HLA-DRB1*04 was more prevalent in the GGA group compared with RO (37.5% vs. 25%). However, this result was not statistically significant (Appendix A). In addition, we independently compared GGA-ILA and RO-ILA vs. HC, finding a similar allele frequency between groups and no statistically significant differences (Appendix A).

## 4. Discussion

ILA represent a complex and still poorly characterized disorder, and the pathophysiological mechanisms implicated in its development and progression are still unclear. A chest HRCT allows the detection of ILA, which are associated with lung function decline and may precede the development of clinically relevant interstitial lung disease (ILD) [17]. In different cohorts, more than 40% of the ILA subjects present progression and an important percentage progress to IPF or other ILD [18].

In this study, we found that ILA subjects were older, mostly smokers with reduced pulmonary function tests. These findings are similar to those reported in other cohorts, such as COPDGene, ECLIPSE, and AGES-Reykjavik, where asymptomatic subjects with ILAs presented poor pulmonary function compared to clinically healthy subjects [2,19,20]. In addition, we compared the subpleural ILA pattern vs. HC, and the results were comparable with the whole ILA group, similar to the previous report by Axelsson et al. [21].

The cellular mechanisms involved in the development of ILA have not been elucidated. In this context, we have recently shown that CD4+ T cells from ILA subjects are highly proliferative and show excessive functional activity [22]. In addition, we found a deregulated cytokine/chemokine profile displayed by CD4+ T cells, suggesting that it may contribute to a higher systemic proinflammatory status and the development of ILA [22].

The HLA complex plays a key role in the genetic control of immune responsiveness. Accordingly, diverse HLA alleles have been associated with an essential number of diseases, such as autoimmune, infectious, tumor immunity, organ transplantation, and reproductive immunity, and a variety of pulmonary diseases that usually show defects in the immune response [23].

In the case of interstitial lung diseases, most associations have implicated the locus HLA-DRB1. Thus, for example, the allele HLA-DRB1*03:01 was found remarkably increased in a subset of patients with hypersensitivity pneumonitis with the presence of circulating autoantibodies, which is also associated with a higher mortality [15]. In the context of autoimmune disease-related ILDs, the HLA-DRB1*14:06, *15, and *16 alleles are associated with a higher risk of developing an ILD in patients with rheumatoid arthritis (RA) in Asian populations, while the HLA-DRB1*04 allele is associated with a reduced risk for ILD [24,25]. Moreover, in about two-thirds of RA patients, the presence of serum antibodies to citrullinated protein antigens is associated with the HLA-DRB1 risk alleles [26]. Similarly, the HLA-DRB1*04:05 allele has been associated with an increased risk for drug-induced ILD in a Japanese population [27].

The HLA-DRB1 gene consists of six exons, each encoding different protein domains. From them, Exon 2 of HLA-DRB1 is the most variable and shares the amino acid sequence of the antigen recognition site [28]. HLA class II molecules mostly sample the extracellular world by presenting antigens captured and processed in endolysosomal compartments. The compartmentalization of proteolysis and the distinct intracellular distributions of HLA molecules are the primary drivers of antigen processing and presentation, which is the primary function of these proteins [29].

In our study (which, to the best of our knowledge, is the first to describe this association), we found that HLA-DRB1*07 is associated with ILAs and the subpleural ILA.

Importantly, in a recent study performed in three Mexican populations, it was found that the HLA-DRB1*07:01 allele is the third most frequent, with a prevalence of 7.37% in all subjects [30], which is markedly lower than we found in the ILA subjects (12.22%).

Outstandingly, when we compared the individuals with subpleural and central ILAs, we found that HLA-DRB1*14 and *15 were more frequent in the central ILA group, suggesting that a different genetic background exists between ILA phenotypes. In agreement with this hypothesis, the *MUC5B* promoter polymorphism is increased in patients with ILA but strongly associated with specific radiologic subtypes of ILA in the AGES-Reykjavik and COPDGene cohorts [8]. Moreover, recently, it was reported that the subpleural subtype of ILA is related to progress radiologically over 4 years [31]. This finding could be related to the differences in allele frequencies between subpleural and central ILAs.

The HLA-DRB1*07 allele has also been associated with sarcoidosis, including in the ILD spectrum. Mirfeizi and coworkers [32] described that patients with sarcoidosis from Northeast Iran have an increased HLA-DRB1*07 allele frequency compared with healthy controls. Additionally, Malkova et al. [33] also described that HLA-DRB1*07, *14, and *15 are associated with chronic sarcoidosis.

This study had limitations. Firstly, our sample size was relatively small and obtained from a single center. Secondly, we did not make a high-resolution genotyping. Thirdly, we did not evaluate other HLA class II regions (DP and DQ) for creating haplotypes.

## 5. Conclusions

In conclusion, the HLA-DRB1*07 allele is associated with ILA, specifically with the subpleural ILA pattern. However, further studies are required to better define the genetic susceptibility to develop ILA and determine the possible progression to established interstitial lung disease.

## Figures and Tables

**Table 1 biomolecules-12-01662-t001:** Demographic and clinical characteristics.

Variables	ILA	HC	*p*-Value	Subpleural	HC	*p*-Value
	(*n* = 90)	(*n* = 189)		(*n* = 75)	(*n* = 186)	
Age, (years)	69 (58–90)	68 (49–87)	0.003	69 (60–90)	68 (49–87)	0.005
Sex, female (%)	60 (66.66%)	117 (62.90%)	0.54	44 (58.66%)	117 (62.90%)	0.52
Smoking Status						
Smoker, yes (%)	55 (61.11%)	88 (47.31%)	0.03	44 (58.66)	88 (47.31%)	0.09
Years of smoking	25 (1–64)	12.5 (0–54)	0.074	20 (1–62)	12.5 (0–54)	0.25
Cigarettes per day	4 (0–40)	4 (0–80)	0.63	4 (0–40)	4 (0–80)	0.73
Tobacco index	5 (0–76)	3 (0–117)	0.133	2 (0–76)	3 (0–117)	
Comorbidities						
DM	18 (20%)	32 (17.20%)	0.57	15 (20%)	32 (17.20%)	0.59
HAS	29 (32.22%)	78 (41.93%)	0.12	22 (29.33)	78 (41.93%)	0.058
Pulmonary function						
6MWT	440 (80–600)	483(104–626)	0.015	460 (80–600)	483(104–626)	0.11
Initial SaO_2_	93 (90–99)	94 (90–99)	<0.001	93 (90–99)	94 (90–99)	0.002
Final SaO_2_	88 (54–97)	93 (67–98)	<0.001	87 (54–96)	93 (67–98)	<0.001
FVC %	92 (49–142)	96 (62–133)	0.016	94.50 (57–142)	96 (62–133)	0.15
FEV1 %	95 (54–147)	102 (62–151)	0.058	100 (68–147)	102 (62–151)	0.26
FEV1/FVC %	78 (61–92)	79 (65–89)	0.96	79 (61–92)	79 (65–89)	0.99
Dlco/VA	95 (57–144)	107 (71–176)	<0.001	95 (57–144)	107 (71–176)	<0.001
Dlco adj	89 (43–116)	98.5 (50–170)	<0.001	90 (43–116)	98.5 (50–170)	<0.001

6MWT: six-minute walking test; DL_CO_: diffusing lung capacity for CO; FEV1: forced expiratory volume in the first second; FVC: forced vital capacity; SAH: systemic arterial hypertension; SaO_2_: oxygen saturation; T2DM: type 2 diabetes mellitus. All values are expressed as median and minimum–maximum values and percentages as appropriate. We used the Mann–Whitney *U* test and Fisher’s exact test. A *p*-value < 0.05 was considered significant.

**Table 2 biomolecules-12-01662-t002:** Allele frequency between the ILA and HC groups.

HLA-DRB1	ILA		HC		*p*-Value	*p*-Value Corrected	OR	95% CI
	*n* = 90	F (%)	*n* = 189	F (%)				
*01	12	6.66	29	7.67	0.65		0.85	0.42–1.71
*03	11	6.11	23	6.08	0		0.99	0.47–2.09
*04	51	28.33	105	27.78	0.92		1.02	0.68–1.51
*07	22	12.22	25	6.61	0.02	0.04	1.95	1.06–3.57
*08	24	13.33	62	16.40	0.33		0.77	0.46–1.29
*09	2	1.11	4	1.06	0.95		1.04	0.18–5.75
*10	2	1.11	4	1.06	0.95		1.04	0.18–5.75
*11	14	7.77	24	6.35	0.54		1.14	0.74–1.77
*12	3	1.66	3	0.79	0.35		2.1	0.42–10.54
*13	7	3.88	20	5.29	0.46		0.72	0.29–1.73
*14	9	5.00	34	8.99	0.09		0.52	0.24–1.12
*15	11	6.11	28	7.41	0.56		0.8	0.39–1.66
*16	12	6.66	17	4.50	0.28		1.5	0.70–3.22

HC: healthy controls group; ILA: interstitial lung abnormalities group. A *p*-value < 0.05 was considered significant.

**Table 3 biomolecules-12-01662-t003:** Logistic regression analysis adjusted for confounding variables in the ILA and HC Groups.

Variables	z-Value	*p*-Value	OR	95% CI
Intercept	2.122	0.034	23.48	1.34–461.27
Age	−3.464	<0.001	0.94	0.90–0.97
Smoking	2.201	0.028	1.82	1.07–3.11
HLA-DRB1*07	2.084	0.037	2.11	1.04–4.28

A *p*-value < 0.05 was considered significant.

**Table 4 biomolecules-12-01662-t004:** Allele frequency between ILA subgroups according to HRCT location.

HLA DRB1	Subpleural ILA		Central ILA		*p*-Value
	*n* = 75	F (%)	*n* = 15	F (%)	
*01	10	6.67	2	6.67	
*03	11	7.33	0	0	
*04	41	27.33	10	33.33	
*07	21	14	1	3.33	
*08	21	14	3	10	
*09	2	1.33	0	0	
*10	2	1.33	0	0	
*11	13	8.67	1	3.33	
*12	2	1.33	1	3.33	
*13	6	4	1	3.33	
*14	5	3.33	4	13.33	0.012
*15	5	3.33	6	20	0.003
*16	11	7.33	1	3.33	

ILA: interstitial lung abnormalities. A *p*-value < 0.05 was considered significant.

**Table 5 biomolecules-12-01662-t005:** Allele frequency between the subpleural ILA and HC groups.

HLA-DRB1	Subpleural ILA		HC		*p*-Value	*p*-Value Corrected	OR	IC 95%
	*n* = 75	F (%)	*n* = 189	F (%)				
*01	10	6.67	29	7.67				
*03	11	7.33	23	6.08				
*04	41	27.33	105	27.78				
*07	21	14	25	6.61	0.007	0.011	2.29	1.24–4.25
*08	21	14	62	16.40				
*09	2	1.33	4	1.06				
*10	2	1.33	4	1.06				
*11	13	8.67	24	6.35				
*12	2	1.33	3	0.79				
*13	6	4.00	20	5.29				
*14	5	3.33	34	8.99				
*15	5	3.33	28	7.41				
*16	11	7.33	17	4.50				

HC: healthy controls group; ILA: interstitial lung abnormalities group. A *p*-value < 0.05 was considered significant.

**Table 6 biomolecules-12-01662-t006:** Logistic regression analysis adjusted for confounding variables in the Subpleural ILA and HC Groups.

Variables	z-Value	*p*-Value	OR	95% CI
Intercept	2.211	0.027	32.66	1.61–769.68
Age	−2.946	0.003	0.94	0.90–0.98
HLA-DRB1*07	2.615	0.009	2.58	1.26–5.28

A *p*-value < 0.05 was considered significant.

## Data Availability

All data generated for this study are included in this article and its Appendix A.

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
