# Peer review of "The HLA-DRB1*07 Allele Is Associated with Interstitial Lung Abnormalities (ILA) and Subpleural Location in a Mexican Mestizo Population"

_biomolecules, 2022, doi:10.3390/biom12111662_

Round 1
Reviewer 1 Report
The authors aimed to evaluate the participation of the HLA class II (HLA-DRB1 loci) in the genetic susceptibility for ILA and the different locations and ILA subtypes in a Mexican mestizo population cohort.
The topic is novel and interesting but some amendments are needed before the paper can be reconsidered.
1. The introduction must be expanded, the background of the study is not solid and accurate as it stands. More relevant references must be included.
2. The discussion section is too superficial in the current format and does not discuss critically the results of the study in the light of the current evidences of the pertinent literature. Please expand this section with more references included.
Thanks for the opportunity to review.
Author Response
The authors aimed to evaluate the participation of the HLA class II (HLA-DRB1 loci) in the genetic susceptibility for ILA and the different locations and ILA subtypes in a Mexican mestizo population cohort.
The topic is novel and interesting but some amendments are needed before the paper can be reconsidered.
- The introduction must be expanded, the background of the study is not solid and accurate as it stands. More relevant references must be included.
Thank you for your comments. Now, the introduction has been improved, reordering and making it more fluid.
- The discussion section is too superficial in the current format and does not discuss critically the results of the study in the light of the current evidences of the pertinent literature. Please expand this section with more references included.
Attending to your kind comment, we have reviewed and improved the discussion and included appropriate references.
Thanks for the opportunity to review.
We really appreciate your time reviewing our manuscript. Your comments have helped us to improve it, thanks to you.
Reviewer 2 Report
Here the authors report differences in in 90 subjects with ILD vs 188 healthy control subject and assess for risk factors for ILD by HLA allele frequency in a Mexican Mestizo Population. This is an important study because most of the genetic association studies in ILD and ILA have reflected patients from the US or Europe. I have a few minor comments:
-Line 30 – “In recent years have been described..” – this sentence contains a type. It should be in recent years ‘it’ has been described…
-Please provide more details as to how the designation and grading of ILA occurred. Were the scans read by 1 or more radiologist? If more than 1, how was consensus achieved if a discrepancy?
-What was the cutoff for subpleural ILA vs central ILA
Author Response
Here the authors report differences in in 90 subjects with ILD vs 188 healthy control subject and assess for risk factors for ILD by HLA allele frequency in a Mexican Mestizo Population. This is an important study because most of the genetic association studies in ILD and ILA have reflected patients from the US or Europe. I have a few minor comments:
-Line 30 – “In recent years have been described..” – this sentence contains a type. It should be in recent years ‘it’ has been described…
Dear reviewer, thank you for your comments; we have reviewed the manuscript and corrected typos and grammar in depth.
-Please provide more details as to how the designation and grading of ILA occurred. Were the scans read by 1 or more radiologist? If more than 1, how was consensus achieved if a discrepancy?
Thank you for your question. Mayra Mejía (MM) and Ivette Buendia-Roldan (IB-R) have a good concordance in evaluating the extent of lung disease in HRCT and have been previously reported (J Rheumatol. 2020;47(3):415-23.). The evaluation of MM was used in the analysis of the data. MM has a good intraobserver agreement (intraclass correlation coefficient 0.90; CI: 0.84 – 0.94).
-What was the cutoff for subpleural ILA vs central ILA
That’s an interesting question; unfortunately, there is no cutoff point for subpleural vs. central ILA; this classification is according to the recommendations of the Fleischner society. Since it is a subjective measurement that radiologists make when evaluating the tomography, it is the predominance of where the lesions are found, and it is not a quantification with software because that has not yet been validated. Now we have included a comment about the Fleischner society at the end of the first paragraph of the material and methods section.
Round 2
Reviewer 1 Report
amended manuscript is acceptable
Author Response
Thanks